# A Quantum Dual-Signature Protocol Based on SNOP States without Trusted Participant

**DOI:** 10.3390/e23101294

**Published:** 2021-09-30

**Authors:** Kejia Zhang, Xu Zhao, Long Zhang, Guojing Tian, Tingting Song

**Affiliations:** 1School of Mathematical Science, Heilongjiang University, Harbin 150080, China; zhangkejia@hlju.edu.cn (K.Z.); 2190972@s.hlju.edu.cn (X.Z.); 2State Key Laboratory of Networking and Switching Technology, Beijing University of Posts and Telecommunications, Beijing 100876, China; 3Laboratory of Cryptology and Cyberspace Security, Heilongjiang University, Harbin 150080, China; 4Institute of Computing Technology, Chinese Academy of Sciences, Beijing 100190, China; tianguojing@ict.ac.cn; 5College of Information Science and Technology, Jinan University, Guangzhou 510632, China; tingtingsong@jnu.edu.cn; 6Guangxi Key Laboratory of Cryptography and Information Security, Guilin 541004, China

**Keywords:** quantum dual-signature, strongly nonlocal orthogonal product states, untrusted third party

## Abstract

Quantum dual-signature means that two signed quantum messages are combined and expected to be sent to two different recipients. A quantum signature requires the cooperation of two verifiers to complete the whole verification process. As an important quantum signature aspect, the trusted third party is introduced to the current protocols, which affects the practicability of the quantum signature protocols. In this paper, we propose a quantum dual-signature protocol without arbitrator and entanglement for the first time. In the proposed protocol, two independent verifiers are introduced, here they may be dishonest but not collaborate. Furthermore, strongly nonlocal orthogonal product states are used to preserve the protocol security, i.e., no one can deny or forge a valid signature, even though some of them conspired. Compared with existing quantum signature protocols, this protocol does not require a trusted third party and entanglement resources.

## 1. Introduction

With the development of the Internet, network communications have become more and more frequent in daily life. Therefore, it is particularly important to improve the security of network communications. Digital signatures are not only widely employed to authenticate the identity of communication participants, but also ensure the integrity of legal messages. For example, encrypting and signing the transmitted messages, verifying the identity of all parties, are necessary to ensure the security of network communications. As we know, the applied classic digital signature protocols are based on computational complexity, such as the decomposition of a large prime number. However, with the development of quantum algorithms [1], classic cryptographic protocols are becoming increasingly insecure. Fortunately, various quantum cryptography protocols, such as quantum key distribution (QKD) [2,3,4], controlled quantum teleportation [5,6,7,8,9,10], quantum secret sharing [11,12,13,14,15] and quantum secure direct communication [16,17,18,19], can be implemented on quantum networks [20,21,22,23,24]. Since their security is based on the laws of quantum mechanics, they are immune to the attacks on quantum computers. During them, designing digital signature protocols based on quantum technology is an important research aspect of quantum cryptographic protocols.

In 2001, Gottesman et al. [25] first proposed a quantum digital signature based on the fundamental principles of quantum physics, i.e., a quantum analogue of one-way functions which required O(m) qubits is used to encrypt an m-bit message. In 2002, Zeng et al. [26] put forward a signature protocol based on the GHZ states, whose realization depends upon a trust arbitrator. Since then, many variations of quantum signature protocols have been presented. For example, Li et al. [27] proposed an arbitrated quantum signature protocol using Bell states instead of GHZ states. Zou et al. [28] presented an arbitrated quantum signature protocol without using entangled states. In 2016, Liu et al. [29] proposed a quantum dual-signature protocol, which combines two signed messages expected to be sent to two different recipients. In their protocol, the entanglement swapping with coherent states is applied. Compared to classic signatures, the quantum signature which involves quantum algorithms, could be more secure and efficient. As we know, quantum signatures also have wide applications in the ecommerce system as classic signatures. 

All these protocols are based on the assumption that a trusted third party exists in the quantum network, though this is not practical. In 2015, Kang et al. [30] first proposed a controlled mutual quantum entity authentication using entanglement swapping by introducing an untrusted third party. However, the controller may perform an internal attack, because the others do not test the correlation of the entangled state. In order to prevent this security loophole, a checking procedure was added to confirm the correlation of the entangled state (see Ref. [31]). Nevertheless, Wang et al. [32] pointed out that there is still a secure flaw in Kang’s improved protocol [31], that is, the untrusted third party can obtain the shared key between the other participants without being detected. In 2016, Li et al. [33] proposed a secure quantum blind dual-signature scheme without arbitrators. It does not rely on an arbitrator in the verification phase as the previous quantum signature schemes do. The security is guaranteed by the entanglement in quantum information processing. Compared with the existing quantum signature protocols, it reduces the over rights of the third party in verification. However, the research on quantum digital signatures without arbitrators is still in its infancy. 

The existing quantum digital signatures without arbitrators are usually implemented by entangled states. As we know, the preparation of entanglement in experiments is difficult. Therefore, it is very urgent to propose a more practical quantum signature protocol without arbitrators. Recently, the expression of quantum nonlocality has attracted more attention. In 2019, Halder et al. [34] first proposed a strong quantum nonlocality without entanglement and presented two explicit strongly nonlocal sets of quantum states in $C^{3}⊗C^{3}⊗C^{3}$ and $C^{4}⊗C^{4}⊗C^{4}$ quantum systems. For the sake of simplicity, we define these states as SNOP states. Compared with normal nonlocal orthogonal product states, SNOP states have strong quantum nonlocality for tripartite, i.e., they are locally irreducible in every bipartition. In this situation, if the private messages are encoded into SNOP states, the security of the private messages is ensured. This means that the attacker cannot determine the accurate whole state even if they obtain two particles of the SNOP states. A secure quantum signature protocol has to meet at least two requirements, i.e., non-forgery and non-repudiation. In this case, we propose a quantum dual-signature protocol based on SNOP states without a trusted third party. The security of our protocol is guaranteed by the secret encryption algorithm [35] and SNOP states. The encryption algorithm is mainly used to resist external attackers. Furthermore, each bipartite of the SNOP states is locally irreducible to resist internal attacks. In this case, the security of our protocol is shown, i.e., neither an external attacker nor an internal one can forge a signature. Moreover, no one can deny the signature. 

The rest of the paper is arranged as follows: In Section 2, some preliminary theories are introduced. In Section 3, we describe the quantum dual-signature protocol including the initializing phase, signing phase and verification phase. The security of our protocol is analyzed in Section 4. Finally, a short conclusion is given in Section 5. 

## 2. Preliminaries

In this section, we describe an encryption algorithm—Key-Controlled-‘I’QOTP. Then, a set of SNOP states to encode messages are introduced.

### 2.1. Key-Controlled-‘I’QOTP 

As we know, the quantum one time pad (QOTP) is an important way to generate quantum signatures [36]. However, Gao et al. [37] pointed out that there exist some security problems in these protocols. In the above security analysis of AQS protocols, one of the most basic assumptions is that the signature is generated by encrypting bitwise messages. In this case, the receiver may forge a legal signature by performing a corresponding operator to the signature and message without secret keys. In 2013, Zhang et al. [35] presented two types of improved encryption algorithms, called Key-Controlled-‘I’QOTP and Key-Controlled-‘T’QOTP, to prevent forgery attacks effectively. Here, we briefly introduce Key-Controlled-‘I’QOTP.

Firstly, a set W with four Clifford operators is introduced to encrypt the message |P〉 to obtain |S〉. Secondly, the two bits $K_{i}$ and $K_{2n-i+1}$ in the shared key string K are appointed to determine the corresponding operator in Table 1. Moreover, the message |P〉 is encrypted into |S〉 in the form of Equation (1).
(1)$$|S\rangle ={⊗}_{i=1}^{n}\sigma_{x}^{k_{2i}}\sigma_{z}^{k_{2i-1}}W_{K_{i}K_{2n-i+1}}|P\rangle$$

Here, we take n=1, i=1, |P〉=α|0〉+β|1〉 and $K_{1}K_{2}=00$ as an example to demonstrate how to encrypt quantum states using secret keys. This is the way to generate quantum signatures in subsequent protocols.
(2)$$\begin{array}{l} |S\rangle ={⊗}_{i=1}^{1}\sigma_{x}^{k_{2}}\sigma_{z}^{k_{1}}W_{K_{1}K_{2}}\left(\alpha |0\rangle +\beta |1\rangle \right)\\ =\sigma_{x}^{0}\sigma_{z}^{0}W_{00}\left(\alpha |0\rangle +\beta |1\rangle \right)\\ =\frac{1}{\sqrt{2}}\left(\left|0\rangle \langle 1|+|1\rangle \langle 0|+|0\rangle \langle 0|-|1\rangle \langle 1\right|\right)\left(\alpha |0\rangle +\beta |1\rangle \right)\\ =\frac{1}{\sqrt{2}}\left(\alpha |1\rangle +\alpha |0\rangle +\beta |0\rangle -\beta |1\rangle \right)\\ =\frac{1}{\sqrt{2}}\left(\left(\alpha +\beta \right)|0\rangle +\left(\alpha -\beta \right)|1\rangle \right) \end{array}$$

Zhang et al. proved that this encryption algorithm can be applied to generate signatures which cannot be forged by the receiver. Therefore, in order to ensure security, Key-Controlled-‘I’QOTP is used in the following quantum dual-signature protocol.

### 2.2. SNOP States 

In order to show our protocol, a type of SNOP states is introduced.
(3)|1〉|2〉|1±2〉,|2〉|1±2〉|1〉,|1±2〉|1〉|2〉|1〉|3〉|1±3〉,|3〉|1±3〉|1〉,|1±3〉|1〉|3〉|2〉|3〉|1±2〉,|3〉|1±2〉|2〉,|1±2〉|2〉|3〉|3〉|2〉|1±3〉,|2〉|1±3〉|3〉,|1±3〉|3〉|2〉|1〉|1〉|1〉,|2〉|2〉|2〉,|3〉|3〉|3〉

In Ref. [34], these states are proved to be locally irreducible in all bipartitions. Since local irreducibility is a sufficient condition for strong nonlocality, these states are strongly nonlocal. In this situation, as the messages are encoded into SNOP states, the security of the private messages is ensured. This means that the attacker cannot determine the accurate forms even if they obtain two particles of the SNOP states.

## 3. Quantum Dual-Signature Protocol with SNOP States

In this section, a quantum dual-signature protocol is given. Firstly, the identity of the participants and the process of the protocol are described in Section 3.1. Then, the specific protocol process is divided into the following three phases: the initializing phase, signing phase and verification phase.

### 3.1. Brief Description

There exist three roles in our protocol:Alice is the applicant and signer;Bob is the first verifier of the message;Charlie is the second verifier.

When the applicant Alice needs to warrant a document, she first signs the application document; then, she sends the document to the first verifier Bob to verify one part of the document; next, Bob sends it to Charlie to verify the other part. The process of this protocol can be briefly seen in Figure 1.

### 3.2. Initializing Phase

Step I1 (Secret Key Assignment): Alice shares the two keys’ sequences $K_{AB}$ and $K_{AC}$ with Bob and Charlie, respectively. Bob shares the key sequence $K_{BC}$ with Charlie. This can be achieved with the quantum key distribution (QKD) technique [2,3,4].

Step I2 (Message Encoding): The message M is divided into n groups, $M=M_{1}||M_{2}||\cdots ||M_{n}$; here $M_{t}$ is a 4-bit of a classical bit sequence, where t=1,2,3,⋯,n. Alice encodes each $M_{t}$ to a quantum sequence |S〉 with the 16 states in Table 2; the remaining 11 states are used to detect eavesdropping in Table 3.

Step I3 (Generating Quantum Sequence): Alice generates three identical sequences |S〉, where the first sequence is denoted by $|S^{a}\rangle$, the second one is $|S^{b}\rangle$ and the last one is $|S^{c}\rangle$. By picking out each particle of above sequences, the corresponding quantum sequences $|S_{1}^{a}\rangle$, $|S_{2}^{a}\rangle$, $|S_{3}^{a}\rangle$, $|S_{1}^{b}\rangle$, $|S_{2}^{b}\rangle$, $|S_{3}^{b}\rangle$, $|S_{1}^{c}\rangle$, $|S_{2}^{c}\rangle$ and $|S_{3}^{c}\rangle$ are generated.

For example, we suppose that $|S\rangle =|\varphi_{1}\rangle |\varphi_{2}\rangle |\varphi_{3}\rangle |\varphi_{4}\rangle |\varphi_{13}\rangle |\varphi_{14}\rangle |\varphi_{15}\rangle |\varphi_{16}\rangle$. Then, the specific forms of $|S^{a}\rangle$, $|S^{b}\rangle$ and $|S^{c}\rangle$ are depicted in Equation (4).
(4)$$\begin{array}{l} |S^{a}\rangle =|1\rangle |1\rangle |2\rangle |3\rangle |1-2\rangle |1-3\rangle |1-2\rangle |1-3\rangle \\ |S^{b}\rangle =|2\rangle |3\rangle |3\rangle |2\rangle |1\rangle |1\rangle |2\rangle |3\rangle \\ |S^{c}\rangle =|1+2\rangle |1+3\rangle |1+2\rangle |1+3\rangle |2\rangle |3\rangle |3\rangle |2\rangle \end{array}$$

### 3.3. Signing Phase

Step S (Sending Sequence to Bob): Firstly, Alice encrypts $|S_{3}^{a}\rangle$ and $|S_{3}^{b}\rangle$ with $K_{AB}$ to obtain $|\bar{S_{3}^{a}}\rangle$ and $|\bar{S_{3}^{b}}\rangle$; then, she encrypts $|S_{1}^{c}\rangle$, $|S_{3}^{c}\rangle$, $|\bar{S_{3}^{a}}\rangle$ and $|\bar{S_{3}^{b}}\rangle$ with $K_{AC}$ to obtain $S_{AC}$.
(5)$$S_{AC}=E_{K_{AC}}\{|S_{1}^{c}\rangle ,|S_{3}^{c}\rangle ,|\bar{S_{3}^{a}}\rangle ,|\bar{S_{3}^{b}}\rangle \}$$

Secondly, Alice encrypts $|S_{2}^{a}\rangle$ and $|S_{2}^{c}\rangle$ with $K_{AC}$ to obtain $|\tilde{S_{2}^{a}}\rangle$ and $|\tilde{S_{2}^{c}}\rangle$; then, she encrypts $|S_{1}^{a}\rangle$, $|S_{1}^{b}\rangle$, $|\tilde{S_{2}^{a}}\rangle$, $|S_{2}^{b}\rangle$, $|\tilde{S_{2}^{c}}\rangle$ and $S_{AC}$ with $K_{AB}$ to obtain $S_{AB}$ as her signature. Finally, she inserts the decoy states randomly in $S_{AB}$ to obtain ${S^{\prime }}_{AB}$ and sends it to Bob. The symbolic representation can be seen in Table 4.
(6)$$S_{AB}=E_{K_{AB}}\{|S_{1}^{a}\rangle ,|S_{1}^{b}\rangle ,|\tilde{S_{2}^{a}}\rangle ,|S_{2}^{b}\rangle ,|\tilde{S_{2}^{c}}\rangle ,S_{AC}\}$$

### 3.4. Verification Phase

Step V1 (Detect Eavesdropping): After Bob announces that he has received the sequences ${S^{\prime }}_{AB}$, Alice tells Bob the positions and the initial states of the decoy particles. Then, Bob measures each of the decoy particles with the corresponding basis and compares the measurement outcome with its initial state to check for eavesdropping. If the error probability is within a certain threshold, Bob recovers the sequences $S_{AB}$; otherwise, he aborts the protocol.

Step V2 ($Bob^{\prime }s\;Verification\;of\;the\;First\;Stage$): Bob decrypts $S_{AB}$ and verifies whether $|S_{1}^{a}\rangle$ is equal to $|S_{1}^{b}\rangle$ with SWAP operations [38,39,40]. Here, $V_{B_{1}}$ represents the results.
(7)$$D_{K_{AB}}\{S_{AB}\}=\{|S_{1}^{a}\rangle ,|S_{1}^{b}\rangle ,|\tilde{S_{2}^{a}}\rangle ,|S_{2}^{b}\rangle ,|\tilde{S_{2}^{c}}\rangle ,S_{AC}\}$$
(8)$$\begin{array}{l} V_{B_{1}}=\left\{\begin{array}{ll} 1, & |S_{1}^{a}\rangle =|S_{1}^{b}\rangle \\ 0, & |S_{1}^{a}\rangle \neq |S_{1}^{b}\rangle \end{array}\right. \end{array}$$

If $V_{B_{1}}=0$, he rejects the quantum signature directly; otherwise, he sends $S_{B}$ to Charlie.
(9)$$S_{B}=E_{K_{BC}}\{|\tilde{S_{2}^{a}}\rangle ,|\tilde{S_{2}^{c}}\rangle ,S_{AC}\}$$

Step V3 ($Charlie^{\prime }s\;Verification\;of\;the\;First\;Stage$): Charlie decrypts $S_{B}$, verifies whether $|S_{2}^{a}\rangle$ is equal to $|S_{2}^{c}\rangle$ and generates $V_{C}$.
(10)$$D_{K_{BC}}\{S_{B}\}=\{|\tilde{S_{2}^{a}}\rangle ,|\tilde{S_{2}^{c}}\rangle ,S_{AC}\}$$
(11)$$D_{K_{AC}}\{|\tilde{S_{2}^{a}}\rangle ,|\tilde{S_{2}^{c}}\rangle \}=\{|S_{2}^{a}\rangle ,|S_{2}^{c}\rangle \}$$
(12)$$\begin{array}{l} V_{C}=\left\{\begin{array}{ll} 1, & |S_{2}^{a}\rangle =|S_{2}^{c}\rangle \\ 0, & |S_{2}^{a}\rangle \neq |S_{2}^{c}\rangle \end{array}\right. \end{array}$$

If $V_{C}$ is equal to 0, Charlie will reject the quantum signature directly; otherwise, he will decrypt $S_{AC}$ and send $S_{C}$ to Bob.
(13)$$D_{K_{AC}}\{S_{AC}\}=\{|S_{1}^{c}\rangle ,|S_{3}^{c}\rangle ,|\bar{S_{3}^{a}}\rangle ,|\bar{S_{3}^{b}}\rangle \}$$
(14)$$S_{C}=E_{K_{BC}}\{|\bar{S_{3}^{a}}\rangle ,|\bar{S_{3}^{b}}\rangle \}$$

Step V4 ($Bob^{\prime }sVerification\;of\;the\;Second\;Stage$): Bob first decrypts $S_{C}$ with $K_{BC}$ and $K_{AB}$; then, he verifies whether $|S_{3}^{a}\rangle$ is equal to $|S_{3}^{b}\rangle$ and generates $V_{B_{2}}$.
(15)$$D_{K_{BC}}\{S_{C}\}=|\bar{S_{3}^{a}}\rangle ,|\bar{S_{3}^{b}}\rangle$$
(16)$$D_{K_{AB}}\{|\bar{S_{3}^{a}}\rangle ,|\bar{S_{3}^{b}}\rangle \}=|S_{3}^{a}\rangle ,|S_{3}^{b}\rangle$$
(17)$$\begin{array}{l} V_{B_{2}}=\left\{\begin{array}{ll} 1, & |S_{3}^{a}\rangle =|S_{3}^{b}\rangle \\ 0, & |S_{3}^{a}\rangle \neq |S_{3}^{b}\rangle \end{array}\right. \end{array}$$

If $V_{B_{2}}$ is equal to 0, Bob will reject the quantum signature directly; otherwise, he will recover the sequence |S〉 by $|S_{1}^{b}\rangle$, $|S_{2}^{b}\rangle$ and $|S_{3}^{b}\rangle$ to obtain $m_{j}^{b}$ according to the rule in Table 5.

Step V5 ($Charlie^{\prime }s\;Verification\;of\;the\;Second\;Stage$): Similarly, Charlie recovers the sequence |S〉 according to $|S_{1}^{c}\rangle$, $|S_{2}^{c}\rangle$ and $|S_{3}^{c}\rangle$, by the rules in Table 5, to obtain $m_{j}^{c}$. If $m_{j}^{b}=m_{j}^{c}$, Charlie will announce it is the valid signature of $m_{j}$; otherwise, he will reject this signature. The process can be briefly seen in Figure 2.

## 4. Security and Efficiency Analysis

As mentioned above, a secure quantum signature protocol has to meet at least two requirements, i.e., non-forgery and non-repudiation. In our protocol, the above requirements are completely satisfied. The security of our protocol is guaranteed by the secret encryption algorithm Key-Controlled-‘I’QOTP and SNOP states. The encryption algorithm is mainly used to resist external attackers. Furthermore, each bipartite of the SNOP states is locally irreducible to resist internal attacks. In this case, the security of our protocol is shown, i.e., neither an external attacker nor an internal one can forge a signature. Moreover, no one can deny the signature. The specific analysis can be seen in the following subsections.

### 4.1. Non-Forgery 

#### 4.1.1. Resistance to Outside Attacks Based on Encryption Algorithm 



$(i)\;Eve\;Forges\;Alice^{\prime }s\;Signature.$



In order to forge Alice’s signature, Eve has to intercept the sequences in Step S. Then, he could change Alice’s signature and replace ${S^{\prime }}_{AB}$ with $S^{\prime \prime }{}_{AB}$. However, since the original keys are shared with QKD, it is impossible for her to succeed. If he attempts to forge Alice’s signature without $K_{AB}$, he will perform an operator corresponding to the signature and message. In our protocol, as Key-Controlled-‘I’QOTP is used to generate signature $S_{AB}$, he would not be able to identify the forms of encryption operators except for Alice and Bob, as shown in Table 6.

From Table 6, it can be seen that, if Eve wants to forge one qubit of Alice’s signature with $\sigma_{x}$ or $\sigma_{y}$, the probability of success will be $\frac{1}{3}$. Moreover, the probability will be $\frac{1}{2}$ if the forgery operation is $\sigma_{z}$. Furthermore, if the length of the message is m, the probability $P_{E}$ of Eve’s forgery’s success will be
(18)$$P_{E}={\left(\frac{1}{3}\right)}^{k}{\left(\frac{1}{2}\right)}^{m-k}$$
where k(0≤k≤m,0≤m≤n) represents the total number of qubits which Eve wants to forge the sequence by $\sigma_{x}$ and $\sigma_{y}$; (m−k) represents the number of qubits forged by $\sigma_{z}$. With this encryption algorithm, Eve can be detected during Bob’s verification phase.



$(ii)\;Eve\;Forges\;Bob^{\prime }s\;Verification\;Results.$



For Eve to intercept the sequences in Step V2, he has to change the sequence $S_{B}$ with ${S^{\prime }}_{B}$ and send it to Charlie. Similarly, Eve cannot know the secret key $K_{BC}$. Therefore, Eve’s forgery will be discovered by Charlie once Eve changes only a small part of $S_{B}$. Similarly, if Eve forges Charlie’s verification results, similarly, his forgery will be found by Bob.

#### 4.1.2. Resistance to Inside Attacks Based on SNOP States 

Above, we guarantee that two verifiers are not dishonest at the same time. This assumption is satisfied with the actual situation. In fact, if both verifiers are dishonest, the protocol will be insecure and impractical. Here, we discuss the dishonesty of the verifier, specifically in the case in which one verifier wants to forge a signature to evade the verification of the other one. 

According to our analysis above, the encryption algorithm can resist some forgery attacks, but, for internal attackers who know the secret key, the security needs to be further discussed as follows. Here, SNOP states provides a nature property for resisting internal attacks, i.e., they are locally irreducible in every bipartition. This means that the attacker cannot determine the accurate whole state even if they obtain two particles of the SNOP states. The specific analysis can be seen in the following subsections.



$(i)\;Bob^{\prime }s\;Forgery.$



This is different from Eve’s forgery of Alice’s signature; Bob does not need to intercept the sequences because he has the secret key $K_{AB}$. If he attempts to replace $S_{AC}$ with ${S^{\prime }}_{AC}$ in Step V2, his forgery will not succeed. Since the sequences $S_{AC}$ are generated by Key-Controlled-‘I’QOTP with the key $K_{AC}$, Bob cannot identify the forms of encryption operators, except for Alice and Charlie. Similar to the analysis above, Bob’s forgery will be discovered by Charlie in Step V3.

Based on the analysis above, we assume that Bob attempts to restore all the SNOP states by the sequences $|S_{1}^{b}\rangle$ and $|S_{2}^{b}\rangle$. He can only choose the measurement basis randomly. There are three possible cases seen in Table 7. The probability that he chooses any basis is $\frac{1}{3}$. From Table 7, we can deduce that the probability of choosing the correct measurement basis and obtain one bit, as shown in Equation (19).


(19)
$$\begin{array}{c} P=\frac{\frac{1}{3}\times 1+\frac{1}{3}\times 2+\frac{1}{3}\times 2+\frac{1}{3}\times 1+\frac{1}{3}\times 1+\frac{1}{3}\times 1+\frac{1}{3}\times 1}{16}=\frac{3}{16} \end{array}$$


For the n length of the quantum sequence, it is not difficult to see that the probability of Bob’s successful attack $P^{\prime }$ tends to be zero with the increase in n in Equation (20).
(20)$$\begin{array}{c} \begin{array}{c} P^{\prime }=P^{n}={\left(\frac{3}{16}\right)}^{n} \end{array} \end{array}$$



$(ii)\;Charlie^{\prime }s\;Forgery.$



We assume Charlie tries to modify $|\bar{S_{3}^{a}}\rangle$ and $|\bar{S_{3}^{b}}\rangle$ according to the sequence $|S_{3}^{c}\rangle$ in his hands to make $|S_{3}^{a}\rangle$ and $|S_{3}^{b}\rangle$ equal. Similarly, Charlie cannot identify the forms of the encryption operators without the key $K_{AB}$. If $|S_{3}^{a}\rangle \neq |S_{3}^{b}\rangle$, Charlie’s forgery will be discovered by Bob in Step V4. The process can be briefly seen in Figure 3.

### 4.2. Non-Repudiation 

Alice could attempt to deny her signature in two ways. One is directly denying her signature. Here Alice’s signatures $S_{AB}$ and $S_{AC}$ are generated by $K_{AB}$ and $K_{AC}$, only known to Bob and Charlie. Once Bob and Charlie have verified the validity of the signatures, she will not be able to deny them. Moreover, since Key-Controlled-‘I’QOTP is used, no one can find the corresponding location without knowing $K_{AB}$ and $K_{AC}$. If $|S_{1}^{a}\rangle =|S_{1}^{b}\rangle$ and $|S_{2}^{a}\rangle =|S_{2}^{c}\rangle$, it will be impossible for Alice to deny.

The other way is that Alice denied the signature after Bob verified it. If $m_{j}^{b}=m_{j}^{c}$, Bob and Charlie will be able to judge whether Alice denies. The sequences $|\bar{S_{3}^{a}}\rangle$ and $|\bar{S_{3}^{b}}\rangle$ are encrypted with the key $K_{AB}$ and the key $K_{AB}$ is generated by Key-Controlled-‘I’QOTP. Therefore, Charlie cannot know the secret key $K_{AB}$. If Charlie attempts to modify the sequence, Bob will find this attack in Step V4. Similarly, Alice will not succeed in denying the signature after Charlie’s verification. 

## 5. Discussion and Conclusions

We summarize and compare our protocol with the quantum signature protocols proposed above in Table 8. Compared with the existing quantum signature protocols, our protocol does not need a trusted third party and entanglement resources for the first time.

Comparing with the existing quantum signature protocols, the signature of our protocol requires the cooperation of two verifiers to complete the whole verification process without an arbitrator. In this case, the function has wide applications in practical management. By introducing SNOP states, the present protocol seems more efficient and easier to be realized in noisy intermediate-scale quantum (NISQ) devices, as no entangled resources are required. According to our analysis, the protocol based on SNOP states is immune to attacks from the inside and outside. In other words, the attacker cannot determine the accurate whole state even if they obtain two particles of the SNOP states. Furthermore, we give a potential application for the SNOP states and put forward a series of ideas. We think SNOP states could also be applied to electronic payments, voting systems and so on. Moreover, we believe that SNOP states must have better application scenarios in the future. Finally, we hope that our results are instructive to further research on other quantum cryptographic protocols.

## Figures and Tables

**Figure 1 entropy-23-01294-f001:**
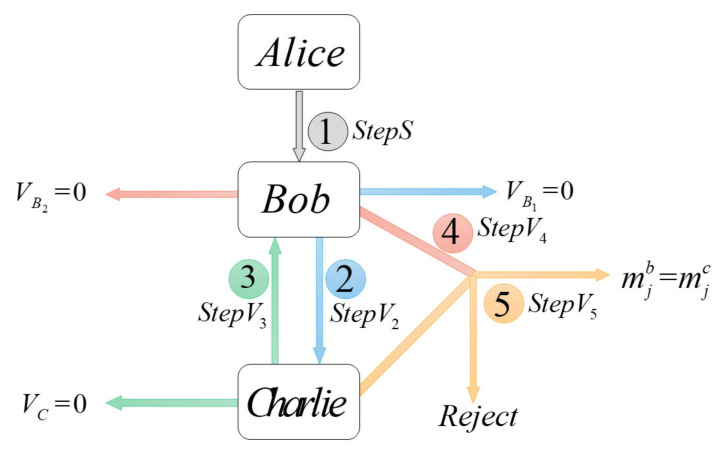
Process of the dual-signature protocol.

**Figure 2 entropy-23-01294-f002:**
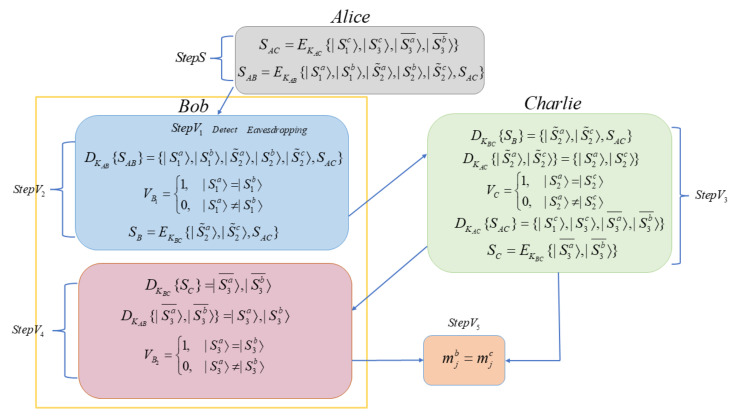
Brief summary of protocol process.

**Figure 3 entropy-23-01294-f003:**
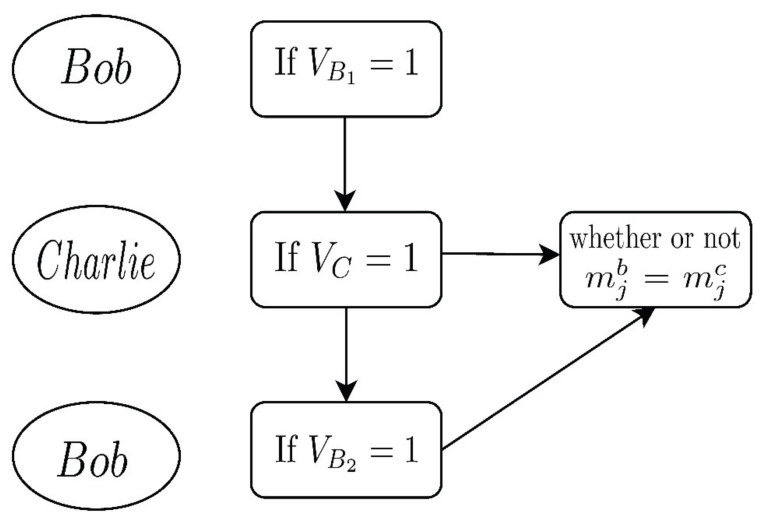
The process of verification phase.

**Table 1 entropy-23-01294-t001:** Corresponding encryption operators in “Key-Controlled-‘I’QOTP”.

$K_{i}K_{2n-i+1}$	Encryption Operator
00	$W_{00}=\frac{1}{\sqrt{2}}(\sigma_{x}+\sigma_{z})$
01	$W_{01}=\frac{1}{\sqrt{2}}(\sigma_{y}+\sigma_{z})$
10	$W_{10}=\frac{1}{2}(I+i\sigma_{x}-i\sigma_{y}+i\sigma_{z})$
11	$W_{11}=\frac{1}{2}(I+i\sigma_{x}+i\sigma_{y}+i\sigma_{z})$

**Table 2 entropy-23-01294-t002:** SNOP states used to encrypt messages.

Message	States	Message	States
$M_{t}=0000$	$|\varphi_{1}\rangle =|1\rangle |2\rangle |1+2\rangle$	$M_{t}=1111$	$|\varphi_{9}\rangle =|1\rangle |2\rangle |1-2\rangle$
$M_{t}=0001$	$|\varphi_{2}\rangle =|1\rangle |3\rangle |1+3\rangle$	$M_{t}=1110$	$|\varphi_{10}\rangle =|1\rangle |3\rangle |1-3\rangle$
$M_{t}=0010$	$|\varphi_{3}\rangle =|2\rangle |3\rangle |1+2\rangle$	$M_{t}=1101$	$|\varphi_{11}\rangle =|2\rangle |3\rangle |1-2\rangle$
$M_{t}=0100$	$|\varphi_{4}\rangle =|3\rangle |2\rangle |1+3\rangle$	$M_{t}=1011$	$|\varphi_{12}\rangle =|3\rangle |2\rangle |1-3\rangle$
$M_{t}=1000$	$|\varphi_{5}\rangle =|1+2\rangle |1\rangle |2\rangle$	$M_{t}=0111$	$|\varphi_{13}\rangle =|1-2\rangle |1\rangle |2\rangle$
$M_{t}=1001$	$|\varphi_{6}\rangle =|1+3\rangle |1\rangle |3\rangle$	$M_{t}=0110$	$|\varphi_{14}\rangle =|1-3\rangle |1\rangle |3\rangle$
$M_{t}=1010$	$|\varphi_{7}\rangle =|1+2\rangle |2\rangle |3\rangle$	$M_{t}=0101$	$|\varphi_{15}\rangle =|1-2\rangle |2\rangle |3\rangle$
$M_{t}=1100$	$|\varphi_{8}\rangle =|1+3\rangle |3\rangle |2\rangle$	$M_{t}=0011$	$|\varphi_{16}\rangle =|1-3\rangle |3\rangle |2\rangle$

**Table 3 entropy-23-01294-t003:** SNOP states used to detect eavesdropping.

Decoy State	
$|\varphi_{17}\rangle =|2\rangle |1+2\rangle |2\rangle$	$|\varphi_{18}\rangle =|2\rangle |1-2\rangle |2\rangle$
$|\varphi_{19}\rangle =|3\rangle |1+3\rangle |1\rangle$	$|\varphi_{20}\rangle =|3\rangle |1-3\rangle |1\rangle$
$|\varphi_{21}\rangle =|3\rangle |1+2\rangle |2\rangle$	$|\varphi_{22}\rangle =|3\rangle |1-2\rangle |2\rangle$
$|\varphi_{23}\rangle =|2\rangle |1+3\rangle |3\rangle$	$|\varphi_{24}\rangle =|2\rangle |1-3\rangle |3\rangle$
$|\varphi_{25}\rangle =|1\rangle |1\rangle |1\rangle$	$|\varphi_{26}\rangle =|2\rangle |2\rangle |2\rangle$
$|\varphi_{27}\rangle =|3\rangle |3\rangle |3\rangle$	

**Table 4 entropy-23-01294-t004:** Symbolic representation.

Sequence	Signature
$|\tilde{S_{2}^{a}}\rangle$	$E_{K_{AC}}\{|S_{2}^{a}\rangle \}$
$|\tilde{S_{2}^{c}}\rangle$	$E_{K_{AC}}\{|S_{2}^{c}\rangle \}$
$|\bar{S_{3}^{a}}\rangle$	$E_{K_{AB}}\{|S_{3}^{a}\rangle \}$
$|\bar{S_{3}^{b}}\rangle$	$E_{K_{AB}}\{|S_{3}^{b}\rangle \}$

**Table 5 entropy-23-01294-t005:** Verifier’s measurement rules.

Message	Bases
$M_{t}=0000\mapsto |\varphi_{1}\rangle =|1\rangle |2\rangle |1+2\rangle$ $M_{t}=1111\mapsto |\varphi_{9}\rangle =|1\rangle |2\rangle |1-2\rangle$	$\{|1\rangle ,|2\rangle ,|3\rangle {\}}_{1}$ $\{|1\rangle ,|2\rangle ,|3\rangle {\}}_{2}$ $\{|1+2\rangle ,|1-2\rangle ,|3\rangle {\}}_{3}$
$M_{t}=0001\mapsto |\varphi_{2}\rangle =|1\rangle |3\rangle |1+3\rangle$ $M_{t}=1110\mapsto |\varphi_{10}\rangle =|1\rangle |3\rangle |1-3\rangle$	$\{|1\rangle ,|2\rangle ,|3\rangle {\}}_{1}$ $\{|1\rangle ,|2\rangle ,|3\rangle {\}}_{2}$ $\{|1+3\rangle ,|1-3\rangle ,|2\rangle {\}}_{3}$
$M_{t}=0010\mapsto |\varphi_{3}\rangle =|2\rangle |3\rangle |1+2\rangle$ $M_{t}=1101\mapsto |\varphi_{11}\rangle =|2\rangle |3\rangle |1-2\rangle$	$\{|1\rangle ,|2\rangle ,|3\rangle {\}}_{1}$ $\{|1\rangle ,|2\rangle ,|3\rangle {\}}_{2}$ $\{|1+2\rangle ,|1-2\rangle ,|3\rangle {\}}_{3}$
$M_{t}=0100\mapsto |\varphi_{4}\rangle =|3\rangle |2\rangle |1+3\rangle$ $M_{t}=1011\mapsto |\varphi_{12}\rangle =|3\rangle |2\rangle |1-3\rangle$	$\{|1\rangle ,|2\rangle ,|3\rangle {\}}_{1}$ $\{|1\rangle ,|2\rangle ,|3\rangle {\}}_{2}$ $\{|1+3\rangle ,|1-3\rangle ,|2\rangle {\}}_{3}$
$M_{t}=1000\mapsto |\varphi_{5}\rangle =|1+2\rangle |1\rangle |2\rangle$ $M_{t}=0111\mapsto |\varphi_{13}\rangle =|1-2\rangle |1\rangle |2\rangle$	$\{|1+2\rangle ,|1-2\rangle ,|3\rangle {\}}_{1}$ $\{|1\rangle ,|2\rangle ,|3\rangle {\}}_{2}$ $\{|1\rangle ,|2\rangle ,|3\rangle {\}}_{3}$
$M_{t}=1001\mapsto |\varphi_{6}\rangle =|1+3\rangle |1\rangle |3\rangle$ $M_{t}=0110\mapsto |\varphi_{14}\rangle =|1-3\rangle |1\rangle |3\rangle$	$\{|1+3\rangle ,|1-3\rangle ,|2\rangle {\}}_{1}$ $\{|1\rangle ,|2\rangle ,|3\rangle {\}}_{2}$ $\{|1\rangle ,|2\rangle ,|3\rangle {\}}_{3}$
$M_{t}=1010\mapsto |\varphi_{7}\rangle =|1+2\rangle |2\rangle |3\rangle$ $M_{t}=0101\mapsto |\varphi_{15}\rangle =|1-2\rangle |2\rangle |3\rangle$	$\{|1+2\rangle ,|1-2\rangle ,|3\rangle {\}}_{1}$ $\{|1\rangle ,|2\rangle ,|3\rangle {\}}_{2}$ $\{|1\rangle ,|2\rangle ,|3\rangle {\}}_{3}$
$M_{t}=1100\mapsto |\varphi_{8}\rangle =|1+3\rangle |3\rangle |2\rangle$ $M_{t}=0011\mapsto |\varphi_{16}\rangle =|1-3\rangle |3\rangle |2\rangle$	$\{|1+3\rangle ,|1-3\rangle ,|2\rangle {\}}_{1}$ $\{|1\rangle ,|2\rangle ,|3\rangle {\}}_{2}$ $\{|1\rangle ,|2\rangle ,|3\rangle {\}}_{3}$

**Table 6 entropy-23-01294-t006:** The attacker’s possible encryption, decryption and possible forgery attacks are under Key-Controlled-‘I’QOTP.

Encryption	Eve’s Possible Decryptions	Eve’s Forgery of Operation	Signature
I	$W_{i}^{+}QW_{i}$		00 01 10 11
$\sigma_{z}$	$W_{i}^{+}\sigma_{z}Q\sigma_{z}W_{i}$	$\sigma_{z}$	$\sigma_{z}$ $\sigma_{x}$ $\sigma_{y}$ $\sigma_{z}$
$\sigma_{x}$	$W_{i}^{+}\sigma_{x}Q\sigma_{x}W_{i}$	$\sigma_{x}$	$\sigma_{y}$ $\sigma_{z}$ $\sigma_{z}$ $\sigma_{x}$
$\sigma_{y}$	$W_{i}^{+}\sigma_{y}Q\sigma_{y}W_{i}$	$\sigma_{y}$	$\sigma_{x}$ $\sigma_{y}$ $\sigma_{x}$ $\sigma_{y}$

**Table 7 entropy-23-01294-t007:** Measurement basis and possible measurement results of attackers.

State	|1〉	|2〉	|3〉	|1+2〉	|1−2〉	|1+3〉	|1−3〉
$A_{1}=\{|1\rangle ,|2\rangle ,|3\rangle \}$	1	1	1	0	0	0	0
$A_{2}=\{|1+2\rangle ,|1-2\rangle ,|3\rangle \}$	0	0	1	1	1	0	0
$A_{3}=\{|1+3\rangle ,|1-3\rangle ,|2\rangle \}$	0	1	0	0	0	1	1

**Table 8 entropy-23-01294-t008:** Comparison among some different quantum signature protocols.

Protocol	Resources	Trust Third Party	Contribution
Zeng et al. [26]	GHZ states	Yes	Arbitrated quantum signature (AQS) protocol is proposed for the first time.
Li et al. [27]	Bell states	Yes	AQS protocol first proposed using Bell states instead of GHZ states.
Zou et al. [28]	Single photon	Yes	AQS protocol without entangled resources is proposed for the first time.
Liu et al. [29]	Coherent states	Yes	Innovation of quantum dual- signature.
Kang et al. [30]	GHZ states	No	The signature protocol without arbitrator is proposed for the first time.
Kang et al. [31]	GHZ-like states	No	Further improvement on the protocol in Ref. [30].
Wang et al. [32]	GHZ-like states, Bell states	No	Further improvement on the protocol in Ref. [31].
Li et al. [33]	Bell states	No	A novel quantum signature protocol without arbitrator is proposed for the first time.
Our protocol	SNOP states	No	A quantum signature protocol without arbitrator and entangled resources is proposed for the first time.

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
