# Peer review of "A Quantum Dual-Signature Protocol Based on SNOP States without Trusted Participant"

_entropy, 2021, doi:10.3390/e23101294_

Round 1

Reviewer 1 Report

.The manuscript discusses a novel quantum authentication protocol which is without a trusted third party for the first time. 

.The manuscript is well organized, the contributions is that it does not need a trusted third party, detail analysis and seem to be new.

.In overall, I accept this manuscript for publication with small revision:

  • Kindly give the comparison with following paper, and insert more detail. https://doi.org/10.1007/s10773-019-04098-4

Author Response

Dear reviewer :

 Thank you for your helpful advice. According to the article you provided, we have modified our paper. The specific modifications are summarized in “Response to reviewer 1”. We hope the revised version could satisfy your requirements.

Best regards,

Xu Zhao et al.

Reviewer 2 Report

I send you the review results of the manuscript titled “A quantum dual-authentication protocol based on SNOP states without trusted participant”. The manuscript ID is entropy-1359303.

The authors propose the quantum dual-authentication protocol based on SNOP state with untrusted third party. The proposed protocol seems to be a quantum signature scheme, not a quantum identity authentication scheme. This is because it is the same mechanism by which an untrusted third party verifies and assures the signer’s signature [1, 2]. Therefore, I highly recommend that the paper is changed to quantum signature using the SNOP state. In particular, considering the security analysis of forgery and non-repudiation, it is proper that this scheme is for quantum signature.

If the title and content are modified to be about quantum signature, this article will be possible to be published. Otherwise, if the authors still want to use the scheme for quantum identity authentication, the security analysis such as impersonation attack and security of the secret key should be performed thoroughly. And then, one more review is necessary.

Other minor comments are as follows:

  1. It is necessary to further describe the advantages of the protocol using the SNOP state.
  2. The author suggested that users perform encryption(E) and decryption(D) when generating S_AB, S_B, and S_C. Here, specific processes for encryption and decryption are required.
  3. Overall, the description of the protocol is insufficient. (e.g., a specific protocol process is required using one of the SNOP states.)

[1] Zeng, Guihua, and Christoph H. Keitel. "Arbitrated quantum-signature scheme." Physical review A 65.4 (2002): 042312.

[2] Li, Qin, Wai Hong Chan, and Dong-Yang Long. "Arbitrated quantum signature scheme using Bell states." Physical Review A 79.5 (2009): 054307.

Author Response

Dear reviewer :

  Thank you very much for your advice. Considering the security analysis of forgery and non repudiation, the protocol is indeed suitable for quantum signature. Therefore, we changed to “A quantum dual-signature protocol based on SNOP states without trusted participant”.  The specific modifications are summarized in “Response to reviewer 2”. We hope the revised version could satisfy your requirements.

Best regards,

Xu Zhao et al.

Reviewer 3 Report

The article is well written and easy to understand. However, few of my feedback can be considered to improve the quality of the paper but all are not necessary.

  1. Introduction may be improved, adding the highlights and the problem statements.
  2. You could improve writing, link better the ideas flow in the Introduction.
  3. Review references because some of them are unstandardized.
  4. The difference between your proposal and related works is not clear, you could do details better. I suggest add a comparative table in ''Related Literature'' to contrast your solution in front of related works.
  5. You could discuss the relationship between your solution and past literature.
  6. Some of the notations are not clear. You can add separate tables for the notation or explain them. Would you please improve the paper formating also?
  7. The paper needs proper proofreading to avoid typos. 

Author Response

Dear reviewer :

 Thank you for your helpful advice. According to your valuable advice, we have modified our paper. The specific modifications are summarized in “Response to reviewer 3”. We hope the revised version could satisfy your requirements.

Best regards,

Xu Zhao et al.

Round 2

Reviewer 2 Report

The authors correspond to my most of comments. I recommend publishing it.